# Rethinking Objectives for Multi-View and Multi-Modal Contrastive Learning

**Panagiotis Koromilas[1], Efthymios Georgiou[2], Giorgos Bouritsas[1,3],**
**Theodoros Giannakopoulos[4], Mihalis A. Nicolaou[5], Yannis Panagakis[1,3]**

[1]**National and Kapodistrian University of Athens**
[2]**ILSP/Athena Research Center**
[3]**Archimedes AI/Athena Research Center**
[4]**NCSR "Demokritos"**
[5]**The Cyprus Institute**

## Abstract

Contrastive Learning (CL), a leading paradigm in Self-Supervised Learning (SSL), typically relies on pairs of data views generated through augmentation. While multiple augmentations per instance (more than two) improve generalization in supervised learning, current CL methods handle additional views suboptimally by simply aggregating different pairwise objectives. This approach suffers from three critical limitations: (L1) it utilizes multiple optimization terms per data point resulting to conflicting objectives, (L2) it fails to model all interactions across views and data points, and (L3) it inherits fundamental limitations (*e.g.* alignment-uniformity coupling) from pairwise CL losses. We address these limitations through two novel loss functions: MV-InfoNCE, which extends InfoNCE to incorporate all possible view interactions simultaneously in one term per data point, and MV-DHEL, which decouples alignment from uniformity across views while scaling interaction complexity with view multiplicity. Our empirical results on ImageNet1K and three other datasets demonstrate that our methods consistently outperform existing multi-view approaches and effectively scale with increasing view multiplicity. We also apply our objectives to multimodal data and show that, in contrast to other contrastive objectives, they can scale beyond just two modalities.
Code: https://github.com/pakoromilas/Multi-View-CL.git

## 1    Introduction

Self-Supervised Learning (SSL) enables learning representations from unlabeled data by exploiting inherent data structure and invariances. Among SSL approaches, Contrastive Learning (CL) has emerged as a leading paradigm by optimizing two complementary objectives: maximizing similarity (alignment) between different views of the same instance while ensuring varying instances remain distinguishable (uniformity / energy) [18]. In this context, *data views refer to variations of the same data point*, such as an image captured from different angles or under different lighting conditions, which can occur naturally in the data or be systematically created *through augmentation* techniques [12].

The benefits of employing multiple views in learning extend beyond mere data augmentation. In supervised representation learning, multiple views per data point improve learning in three key ways: (i) enabling higher learning rates through more stable gradient updates [6], (ii) accelerating convergence by reducing gradient variance [7], and (iii) improving out-of-distribution generalization through implicit regularization [10]. Recognizing these advantages, several SSL methods have adopted multiple views: SwAV [3] employs multi-resolution crops for clustering-based learning, DINO [4] combines global and local views for knowledge distillation, and VICRegL [2] enforces

multi-scale consistency in representation learning. Each method demonstrates that incorporating *more than two views* improves representation quality and downstream performance.

Current multi-view CL methods solely aggregate pairwise losses [17, 15], an approach fundamentally limited in four critical aspects. **Multi-term optimization (L1)**: In current objectives increasing view multiplicity also increases the number of loss terms per instance (one for each view), forcing each representation to satisfy multiple potentially conflicting objectives. **Missing concurrent interactions across all views (L2)**: Although similarity measures are employed to guide optimization across views and instances, current objectives fail to capture all possible interactions among views within a batch thus not guaranting optimization of the desideratum which requires simultaneously aligning all n-views, and contrasting each single-view datapoint to all views of the remaining datapoints in the batch. **Alignment-uniformity coupling (L3)**: Each pairwise comparison inherits fundamental CL issues where view interactions contribute to both alignment and uniformity calculations, resulting in conflicting objectives [8] that worsen as view multiplicity increases.

To achieve the benefits of multiple views in Contrastive Learning and address the limitations of current methods, we design objectives based on three fundamental principles. First, given a data point $i$ and a view of interest $l$, (P1) simultaneous alignment requires all other views of the same data point to be simultaneously aligned, ensuring invariance to all transformations without competing objectives. Second, (P2) accurate energy term mandates that the uniformity component must capture all pairwise interactions in the representation configuration. Third, (P3) one term per data instance maintains a single optimization term per instance for the complete objective, ensuring better optimization. Current objectives violate all three principles, resulting in the three limitations (L1-3) identified above.

Guided by these principles, we introduce two novel multi-view contrastive objectives that address the fundamental limitations of existing approaches:

1. **MV-InfoNCE**: We generalize InfoNCE to capture interactions between all views simultaneously, rather than just summing pair-wise losses, in a single term adressing both (L1) and (L2).

2. **MV-DHEL**: We extend the DHEL [8] loss to decouple alignment from uniformity across views and enable richer interactions that scale with the number of views. This resolves the alignment-uniformity coupling (L3) that becomes more severe as view multiplicity increases.

Empirically, our methods demonstrate three key advantages. First, they achieve *higher downstream accuracy* scores across multiple datasets. Second, they show improved *scalability with increasing number of views*. Finally, although designed for single-view unimodal learning, we demonstrate that, unlike existing contrastive methods [13], which struggle to generalize beyond two modalities [14, 11, 16], our approach is *effective in multimodal settings* that extend beyond the typical two-modality framework [13].

## 2 Multi-View Contrastive Objectives

### 2.1 Notation

Vectors and matrices are denoted by lowercase and uppercase bold letters respectively, $\mathbf{u}$, $\mathbf{U}$, tensors are represented by uppercase bold upright letters $\mathbf{U}$ and sets with calligraphic letters $\mathcal{U}$. An element (scalar) within a matrix/tensor $\mathbf{U}$ is accessed using subscript notation, such as $\mathbf{U}_{i,j,k}$. Fibers (generalisation of rows and columns from matrices to tensors) are represented by fixing all indices except one. For instance, mode-1 fibers of $\mathbf{U}$ are denoted by $\mathbf{U}_{:,j,k}$. Similarly, slices (matrices) of a tensor are formed by fixing one index, *i.e.* $\mathbf{U}_{i,:,:}$. To denote vertical (row-wise) concatenation of matrices $\mathbf{X}$ and $\mathbf{Y}$, we use $[\mathbf{X}; \mathbf{Y}]$, while for depth stacking, i.e., combining matrices as slices of a tensor along a new dimension, we use $[\mathbf{X}, \mathbf{Y}]$. Further, we denote with $[N]$ the set of indices $\{1, ..., N\}$ with cardinality N. In the **multi-view learning setup**, each data point is represented by a collection of $N$ different views, $\mathbf{X} = [\mathbf{x}_1; \ldots; \mathbf{x}_N]$ and $\mathbf{X} = [\mathbf{X}_1, \ldots, \mathbf{X}_M]$ denotes a collection (batch) of $M$ input data points. The corresponding set of $M$ representations is given by $\mathbf{U} = [\mathbf{U}_1, \ldots, \mathbf{U}_M] \in \mathbb{R}^{M \times N \times d}$, where $\mathbf{U}$ equals $f_{\boldsymbol{\theta}}(\mathbf{X}) = [f_{\boldsymbol{\theta}}(\mathbf{x}_1); \ldots; f_{\boldsymbol{\theta}}(\mathbf{x}_M)]$.

Table 1: Comparison of multi-view contrastive objectives for $M$ instances and $N$ views based on three principles: (P1) simultaneous alignment of all views, (P2) accurate energy term with complete pairwise interactions, and (P3) one term per instance. Only MV-InfoNCE and MV-DHEL satisfy all principles. MV-DHEL has the smaller complexity while its the only one that decouples the optimisation of alignment and uniformity.

| Method | Objective | Complexity | P1: Simultaneous Alignment | P2: Accurate Energy Term | P3: Loss Terms per Instance | Decoupled Alignment-Uniformity |
|---|---|---|---|---|---|---|
| pwe | $\dfrac{2}{N(N-1)M} \displaystyle\sum_{\substack{l\in[N] \\ m\in[N],m>l \\ i\in[M]}} \log\left(\dfrac{e^{\mathbf{u}_{i,l}^\top \mathbf{u}_{i,m}/\tau}}{\sum_{j\in[M]} e^{\mathbf{u}_{i,l}^\top \mathbf{u}_{j,m}/\tau}}\right)$ | $\mathcal{O}(M^2N^2)$ | ✗ | ✗ | $\frac{1}{2}N(N-1)$ | ✗ |
| PVC | $\dfrac{-1}{(N-1)M} \displaystyle\sum_{\substack{l\in[N] \\ l'\in[N]\backslash l \\ i\in[M]}} \log\left(\dfrac{e^{\mathbf{u}_{i,l}^\top \mathbf{u}_{i,l'}/\tau}}{e^{\mathbf{u}_{i,l}^\top \mathbf{u}_{i,l'}/\tau} + \sum_{\substack{m\in[N] \\ j\in[M]\backslash i}} e^{\mathbf{u}_{i,l}^\top \mathbf{u}_{j,m}/\tau}}\right)$ | $\mathcal{O}(M^2N^3)$ | ✗ | ✗ | $N(N-1)$ | ✗ |
| MV-InfoNCE | $\dfrac{1}{M}\displaystyle\sum_{i=1}^M \log\left(\dfrac{\sum_{\substack{l\in[N] \\ l'\in[N]\backslash l}} e^{\mathbf{u}_{i,l}^\top \mathbf{u}_{i,l'}/\tau}}{\sum_{\substack{l\in[N]\backslash l \\ m\in[N] \\ j\in[M]}} e^{\mathbf{u}_{i,l}^\top \mathbf{u}_{j,m}/\tau}}\right)$ | $\mathcal{O}(M^2N^2)$ | ✓ | ✓ | 1 | ✗ |
| MV-DHEL | $\dfrac{1}{M}\displaystyle\sum_{i=1}^M \log\left(\dfrac{\sum_{\substack{l\in[N] \\ l'\in[N]\backslash l}} e^{\mathbf{u}_{i,l}^\top \mathbf{u}_{i,l'}/\tau}}{\prod_{l\in[N]}\sum_{j\in[M]} e^{\mathbf{u}_{i,l}^\top \mathbf{u}_{j,l}/\tau}}\right)$ | $\mathcal{O}(M^2N)$ | ✓ | ✓ | 1 | ✓ |

## 2.2 Aggregating Two-View Losses

The common approach in the literature when extending to more than two views is to perform different types of aggregation of pair-wise objectives. Typical examples are **pwe** and **PVC** in Table 1. Beyond the inherent limitations of pairwise loss aggregation, which **precludes direct interaction among all views** and fails to **simultaneously align all views** of the same data point, a more fundamental challenge arises in multi-view contrastive learning: **each representation must simultaneously satisfy multiple, potentially conflicting objectives**.

## 2.3 Principles for Multi-View Contrastive Objectives

To properly extend contrastive losses to multiple views, we define three fundamental principles that must be satisfied for a theoretically sound multi-view contrastive objective:

**P1: Simultaneous Alignment**. Given a data point $i$ and a view of interest $l$, all other views of the same data point must be simultaneously aligned within a single term of the objective. This ensures that representations become *invariant to all transformations* simultaneously, avoiding competing optimization terms that could lead to suboptimal solutions or training instability.

**P2: Accurate Energy Term**. Contrastive objectives fundamentally optimize for alignment and uniformity [18]. The uniformity term corresponds to minimizing the energy of a point configuration $\{\mathbf{u}_1, \ldots, \mathbf{u}_M\}$, a set of representations in our case. Specifically, we seek to minimize the total pairwise energy $\sum_i \sum_j K(\mathbf{u}_i, \mathbf{u}_j)$ [8], where $K$ is typically a gaussian kernel. As shown in [18], this energy minimization is equivalent to minimizing $\sum_i \log \sum_j K(\mathbf{u}_i, \mathbf{u}_j)$, which forms the uniformity term. To maintain theoretical consistency with contrastive objectives, the negative set used in uniformity calculations must represent a complete point configuration that captures *all possible pairwise interactions*.

**P3: One Term per Data Instance**. Current multi-view contrastive losses utilize multiple optimization terms per instance, introducing competing objectives for the same data point. A principled extension to multiple views should maintain one term per instance for the complete objective (*i.e.*, alignment + uniformity), consistent with two-view objectives, to ensure stable and efficient optimization.

Based on these principles we define **MV-InfoNCE** and **MV-DHEL** as presented in Table 1.

Table 2: Linear probing based performance comparison with accuracy and improvement (Diff) based on using one lesser view for each method. Green values indicate the best performance per dataset, bold values indicate the highest values per view.

| Dataset | # Views | pwe | | avg | | PVC | | MV-InfoNCE | | MV-DHEL | |
|---|---|---|---|---|---|---|---|---|---|---|---|
| | | Accuracy | Diff | Accuracy | Diff | Accuracy | Diff | Accuracy | Diff | Accuracy | Diff |
| | 2 | 86 | – | 86 | – | 85.8 | – | 86 | – | **87.4** | – |
| **CIFAR10** | 3 | 87.5 | +1.5 | 87.4 | +1.4 | 87.0 | +1.2 | 87.8 | **+1.8** | **89.1** | +1.7 |
| | 4 | 88.7 | +1.2 | 88.2 | +0.8 | 88.0 | **+1.0** | 88.8 | **+1.0** | 89.5 | +0.4 |
| | 2 | 58.1 | – | 58.1 | – | 57.3 | – | 58.1 | – | **59.4** | – |
| **CIFAR100** | 3 | 59.9 | +1.8 | 60.4 | +2.3 | 60.3 | +3.0 | 60.6 | **+2.5** | **61.8** | +2.4 |
| | 4 | 60.9 | +1.0 | 60.8 | +0.4 | 61.1 | +0.8 | 61.2 | +0.6 | 62.7 | **+0.9** |
| | 2 | 72.2 | – | 72.2 | – | 72.2 | – | 72.2 | – | **73.3** | – |
| **ImageNet-100** | 3 | 75 | +2.8 | 74.8 | +2.6 | 75 | +2.8 | 75.2 | +3 | **77.1** | **+3.8** |
| | 4 | 73.9 | -1.1 | 73.7 | -1.1 | 74.4 | -0.6 | 75.8 | **+0.6** | 77.2 | +0.1 |
| | 2 | 60 | – | 60 | – | 59.7 | – | 60 | – | **61.2** | – |
| **ImageNet-1K** | 3 | 61.2 | +1.2 | 61 | +1 | 61.4 | **+1.7** | 60.8 | +0.8 | **61.9** | + 0.7 |
| | 4 | 62 | + 0.8 | 61.6 | +0.6 | 62.4 | **+0.7** | 61.2 | + 0.4 | 62.6 | + 0.7 |

# 3 Experimental Evaluation

## 3.1 Downstream Performance

We empirically validate our proposed objectives (MV-InfoNCE and MV-DHEL) by benchmarking against three established techniques. These methods are as follows: (i) **pwe** — aggregation based on the pairwise loss computed for all pairs of views; (ii) **avg** — pairwise loss between each view and the mean vector of the remaining views; and (iii) **PVC** as proposed in [15]. Experiments are conducted on four standard image classification datasets: *CIFAR10, CIFAR100, ImageNet-100, and ImageNet1K*, following common SSL practices [19, 20, 22, 18, 5]. We use ResNet50 for ImageNet-100/ImageNet1K and ResNet18 for CIFAR10/CIFAR100. Models are trained for 100 epochs on ImageNet1K (batch size 512, SGD optimizer) and 200 epochs on other datasets (batch size 256, SGD).

In Table 2, we compare the performance of linear probing for all five methods across four datasets under varying view multiplicities (2, 3, and 4). Our results demonstrate the clear advantages of our proposed approaches. **MV-DHEL** consistently achieves the **highest overall accuracy** across all datasets, while **MV-InfoNCE** exhibits the greatest **performance scaling** as views increase. Unlike baseline methods, both our approaches show significant accuracy gains when advancing from two to four views, confirming their effectiveness in leveraging multi-view information.

## 3.2 Application to Multimodal Data

Table 3: Performance comparison on multimodal sentiment analysis datasets. The supervised method provides an upper bound reference. Values in parentheses indicate absolute improvement over the best baseline.

| Dataset | Metric | Methods | | | | | |
|---|---|---|---|---|---|---|---|
| | | Supervised | PWE | AVG | PVC | MV-InfoNCE | MV-DHEL |
| **CMU-MOSEI** | **Accuracy**(↑) | 82.9 | 75.4 | 75.7 | 74.3 | 76.8 (+1.1) | **79.6** (+3.9) |
| | **MAE**(↓) | 0.587 | 0.707 | 0.713 | 0.755 | 0.708 (-0.001) | **0.668** (-0.039) |
| **CH-SIMS** | **Accuracy**(↑) | 83.2 | 76.1 | 76.3 | 68.1 | 76.6 (+0.3) | **79.4** (+3.1) |
| | **MAE**(↓) | 0.354 | 0.448 | 0.468 | 0.563 | 0.421 (-0.027) | **0.392** (-0.056) |

Unlike views that share the same underlying distribution, modalities originate from distinct ones. By using separate encoders for each modality, their representations can be treated as alternative views of the same data point, allowing direct application of contrastive learning to multimodal data (e.g., CLIP [13]). However, CL methods are mainly designed for bimodal settings and struggle with scaling to multiple modalities [14, 11, 16]. Here we empirically evaluate the effectiveness of our method to multimodal setups. We apply our methods on two datasets ( CMU-MOSEI [1] and CH-SIMS [21]) to Multimodal Sentiment Analysis (MSA), a well-established multimodal task [9] that integrates three heterogeneous modalities: audio, vision, and text.

Table 3 demonstrates that, again, **MV-DHEL performs significantly higher** in the multimodal setup, greatly outperforming other methods. The simultaneous utilization of all multimodal interactions places MV-InfoNCE second, while pairwise loss aggregations (pwe and avg) yield slightly lower performance. An interesting observation is that PVC performs notably worse, suggesting it is not suited for multimodal learning.

## 4   Conclusion

We presented a principled approach to multi-view CL through two theoretically grounded objectives: MV-InfoNCE and MV-DHEL. Unlike current methods that process views through pair-wise loss aggregations, our framework enables interactions between all views simultaneously. Concretely, MV-InfoNCE generalises InfoNCE to handle multiple views, while MV-DHEL further addresses alignment-uniformity coupling. Our extensive experiments demonstrate key advantages: (i) improved downstream accuracy, (ii) better scalability as the number of views increases, and (iii) multimodal applicability. Our implementation can be found in https://github.com/pakoromilas/Multi-View-CL.git

## Acknowledgments and Disclosure of Funding

Panagiotis Koromilas was supported by the Hellenic Foundation for Research and Innovation (HFRI) under the 4th Call for HFRI PhD Fellowships (Fellowship Number: 10816). Giorgos Bouritsas and Yannis Panagakis were supported by the project MIS 5154714 of the National Recovery and Resilience Plan Greece 2.0 funded by the European Union under the NextGenerationEU Program. Theodoros Giannakopoulos was supported by the European High-Performance computing Joint Undertaking (JU) under grant agreement No 101234269 and the Greek Ministry of Digital Governance. Mihalis Nicolaou was supported by the TensorICE project (EXCELLENCE/0524/0407), implemented under the social cohesion programme "THALIA 2021-2027", co-funded by the European Union through the Research and Innovation Foundation. This research was partially supported by a grant from The Cyprus Institute on Cyclone.

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
