# OpenReview forum: "Rethinking Objectives for Multi-View and Multi-Modal Contrastive Learning"
_NeurIPS.cc/2025/Workshop/UniReps — UniReps2025_

### Official Review · Reviewer_3oGJ · 2025-09-03
**MV-InfoNCE & MV-DHEL Review**

**Confidence:** 4

**Review:**

*Summary:* The paper proposes two new objectives for multi-view contrastive learning: MV-InfoNCE, which unifies all view interactions in a single per-instance term, and MV-DHEL, which decouples alignment from uniformity across views. These are motivated by three principles (simultaneous alignment, accurate energy, one term per instance) and aim to overcome the limitations of pairwise losses.

*Strengths:*
- Identifies concrete shortcomings of pairwise multi-view objectives.
- Grounding in P1–P3 provides a structured way to frame multi-view CL objectives.
- Both MV-InfoNCE and MV-DHEL are logical, nontrivial extensions of standard losses.
- Demonstrates consistent improvements across multiple datasets and with increasing number of views.
- Application beyond two modalities is a valuable extension.

*Weaknesses:*
- Unlike P1 and P2, which map directly to known desiderata of contrastive learning (alignment + uniformity), P3 (“one term per instance”) seems more like a design heuristic. Multiple objectives per instance can sometimes help by providing complementary regularization. The paper may overstate its theoretical necessity.
- While SwAV, DINO, VICRegL are mentioned, the structural relationship between their objectives and the proposed ones is not explored in depth.
- Claiming scalability with views is attractive, but a more explicit analysis of computational overhead (vs. pairwise methods) is missing.

**Score:**

4

**Topic Fit:**

3

---

### Official Review · Reviewer_E67h · 2025-09-05
**Review of “Rethinking Objectives for Multi-View and Multi-Modal Contrastive Learning”**

**Confidence:** 4

**Review:**

This paper addresses the limitations of extending contrastive learning to multiple views by moving beyond simple aggregation of pairwise losses. The authors introduce two novel objectives, MV-InfoNCE and MV-DHEL. They report improvements across CIFAR, ImageNet subsets, and multimodal sentiment analysis benchmarks. Overall, the paper raises an important issue: naïve aggregation of pairwise objectives in multi-view contrastive learning is suboptimal. The proposed alternatives are sensible extensions, and the empirical results show promise. However, the conceptual framework (L1–L3 vs. P1–P3) feels somewhat circular and repetitive, and the lack of a supervised multi-view baseline or deeper theoretical justification limits the strength of the contribution. Moreover, important experimental details are missing, and the writing could be tightened considerably. I believe an improvement in the clarity of the text would be necessary.


**Strengths**

Multi-view and multimodal extensions of contrastive learning are generally limited in number of modalities, and the paper highlights key shortcomings of existing pairwise aggregation approaches.

Clarity of motivation: The identification of limitations (L1–L3) provides a clean setup for introducing new objectives.

Empirical results: Results span multiple datasets (CIFAR, ImageNet, CMU-MOSEI, CH-SIMS), showing consistency.

**Weaknesses and Concerns**

Overlap and redundancy in principles/limitations:

L1 (“multiple optimization terms per instance”) and L3 (“alignment–uniformity coupling”) substantially overlap; similarly, P1–P3 map almost one-to-one onto L1–L3. The distinctions are not crisp, and at times the text repeats the same point with different labels. This reduces the conceptual neatness claimed. Exaplanations of P1-3 are repeated, while no conclusion/discussion section is present. I believe the authors could do a better structuring of their work.

The justification for exactly these three principles feels more like design choices than theoretically necessary conditions. Why only these and not others? The paper frames P1–P3 as “fundamental principles,” but offers little in the way of proof or deeper justification. They seem motivated by convenience rather than necessity. Without a stronger theoretical analysis, the claims risk sounding prescriptive.

Baselines and comparisons:

Results omit a supervised multi-view baseline. Since multiple views are well-studied in supervised learning, such a baseline would provide a stronger reference point for improvements, similarly to the multimodality setting.

Experimental detail gaps:

It is not explained how additional views are generated for CIFAR and ImageNet. Are these different augmentations, multi-crops, or something else? This detail is crucial to reproduce and interpret the results.

The comparison between MV-InfoNCE and MV-DHEL would benefit from a deeper discussion. They differ primarily in the denominator in their equations, yet the interpretation/intutition of why one outperforms the other is missing.

Writing and presentation:

The paper ends abruptly after experiments. A proper conclusion/discussion section is missing. The reader is left without synthesis or guidance on limitations, future work, or broader implications.


Recommendation: Weak Reject — the contribution is interesting and could spark discussion at a workshop, but in its current form the paper lacks the rigor, clarity, and completeness expected. Strengthening the theoretical justification, clarifying experimental details (view generation, denominator interpretation), and adding a conclusion would improve it significantly.

**Score:**

2

**Topic Fit:**

3

---

### Official Review · Reviewer_tLq4 · 2025-09-17
**Review of submission 146**

**Confidence:** 5

**Review:**

The paper present two novel techniques for contrastive learning aiming at resolving weaknesses of previous state-of-the-art baselines.
The approach is principled and favors creating non-contrasting loss objectives for each data point and augmentation, resulting in better overall scores in both image data and multi-modal sentiment analysis. The paper is very interesting and the application to multi-modal data can be expanded, potentially leading to a strong conference submission.

GIven the short lenght of the paper, I struggled to understand some details of the methods. In fact, given that both losses are small in the text and the intuition on the exponential terms is missing, it is not straightforward to see how new losses incentivizes desired alignment and uniformity, vs limitations of other competitors.

It would be interesting to investigate how these principles, can be applied to CLIP like objectives (which is contrastive but basically enforces alignment between image and captions) and it would be interesting to see if this can improve scaling and training of foundation models.

**Score:**

5

**Topic Fit:**

3